# Unlocking the general relationship between energy and entanglement spectra via the wormhole effect

Zheng Yan [1,2,3] ✉ & Zi Yang Meng [1] ✉

Based on the path integral formulation of the reduced density matrix, we develop a scheme to overcome the exponential growth of computational complexity in reliably extracting low-lying entanglement spectrum from quantum Monte Carlo simulations. We test the method on the Heisenberg spin ladder with long entangled boundary between two chains and the results support the Li and Haldane's conjecture on entanglement spectrum of topological phase. We then explain the conjecture via the wormhole effect in the path integral and show that it can be further generalized for systems beyond gapped topological phases. Our further simulation results on the bilayer antiferromagnetic Heisenberg model with 2D entangled boundary across the $(2+1)$D O(3) quantum phase transition clearly demonstrate the correctness of the wormhole picture. Finally, we state that since the wormhole effect amplifies the bulk energy gap by a factor of $\beta$, the relative strength of that with respect to the edge energy gap will determine the behavior of low-lying entanglement spectrum of the system.

The fruitful dialog and fusion between quantum informatics and highly entangled condensed matter systems, have been gradually appreciated and recognized in recent years[1,2]. Within this trend, quantum entanglement serves as the quintessential quantity to detect and characterize the informational, field-theoretical and topological properties of many-body quantum states[3–6]. It offers, among many interesting features, the direct connection to the conformal field theory (CFT) and categorical description of the problem at hand[7–24]. More than a decade ago, Li and Haldane proposed that the entanglement spectrum (ES) is an important, maybe more fundemental, measurement in this regard than the entanglement entropy (EE)[25–27]. Although the generality of such statement has been questioned[28], since then, low-lying ES has been nevertheless widely employed/discussed as a fingerprint of CFT and topology in the investigation in highly entangled quantum matter[29–50]. Moreover, for topological states (e.g. quantum Hall state), they pointed out a possible deep correspondence between the low-lying ES and the true energy spectra on the edge. This

is another famous Haldane's conjecture, other than the one for the gapped spin-1 chain. Later, Qi, Katsura and Ludwig theoretically demonstrated the general relationship between entanglement spectrum of $(2+1)$D gapped topological states and the spectrum on their $(1+1)$D edges[51]. However, besides such gapped topological phases, how universal the Li and Haldane's conjecture is remains an interesting and open question to this day.

On the numeric front, most of the ES studies so far have focused on (quasi) 1D systems. Due to the exponential growth of computation complexity and memory cost, the existing numerical methods such as exact diagonalization (ED) and density matrix renormalization group (DMRG) have obvious limitations for entangling region with long boundaries and higher dimensions. Quantum Monte Carlo (QMC) on the other hand is a powerful tool for studying large size and higher dimensional quantum many-body systems, as the importance sampling scheme can in principle convert the exponential complexity into polynomial when there exists a sign bound for the Hamiltonian

[1]Department of Physics and HKU-UCAS Joint Institute of Theoretical and Computational Physics, The University of Hong Kong, Pokfulam Road, Hong Kong SAR, China. [2]Department of Physics, School of Science, Westlake University, 600 Dunyu Road, Hangzhou 310030 Zhejiang Province, China. [3]Institute of Natural Sciences, Westlake Institute for Advanced Study, 18 Shilongshan Road, Hangzhou 310024 Zhejiang Province, China. ✉e-mail: zhengyan@westlake.edu.cn; zymeng@hku.hk

simulated[52,53]. But in the first appearance, it looks difficult to obtain ES from QMC, as it can't obtain the quantum (ground state) wavefunction directly. However, it has been successfully shown that the computation of Rényi EE can be cast into the sampling of the partition function in modified manifold with different boundary condition for the entangling region and the rest of the system[2,54–60], and the universal information of many interesting quantum many-body phases and phase transitions in EE have been reliably extracted in QMC simulations[17,19,21,61–65].

This paper is a response to these open questions and recent developments. Here we develop a protocol to overcome the exponential growth of computational complexity in obtaining the ES via QMC combined with stochastic analytic continuation[66–76]. Then, we test the method on a Heisenberg ladder with long entangling region between two coupled chains. Our QMC results remove the finite size effects in previous ED results[27] and further show the low-lying ES does indeed behave as the energy spectra of one chain, highly consistent with Handane's conjecture. In the next step, the principle of equivalence is proposed based on the rules of worldline evolution to explain the similarity of ES and edge energy spectrum for this ladder system. According to such physical picture, Handane's conjecture should not only be true in the gapped topological phase, but can be further generalized to other systems such as the quantum critical point (QCP), as long as the subsystem $A$ has only edge without bulk. We then compute the ES of antiferromagentic (AFM) Heisenberg bilayer model across the $(2+1)$D O(3) QCP, where the entanglement boundary is the entire 2D layer, to demonstrate the correctness of the equivalence principle. Furthermore, to extend the understanding of Haldane's conjecture for the general subsystem with both edge and bulk, we develop a pedagogical theory of phenomenology within the path-integral formulation to explain Handane's conjecture, in which we find a *wormhole* effect on the imaginary time edges of environment to induce the modes of entanglement Hamiltonian (EH). Since the wormhole effect amplifies the bulk energy gap by a factor of $\beta = \frac{1}{T}$, it is the relative strength of that with respect to the edge energy gap that will determine the behavior of low-lying ES of the system. Therefore, the bulk gap becomes much larger thus the edge gap contributes almost all the low-lying ES at low temperature $\beta \to \infty$. The wormhole mechanism is more general than the Li and Haldane conjecture in that, it not only explains the working of the conjecture, but also has predicting power to suggest situation where the ES is different from the energy spectrum on the edge but resembles that in the bulk[77].

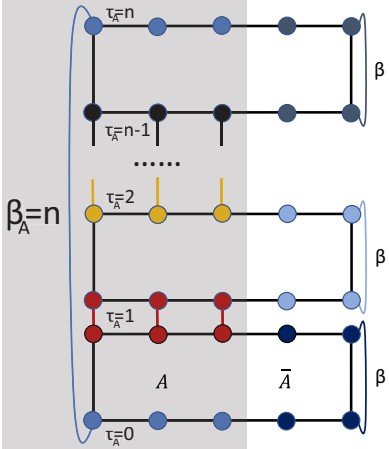

**Fig. 1 | A geometrical presentation of the partition function $\mathcal{Z}_A^{(n)}$.** The entangling region $A$ bewteen replicas is glued together in the replica imaginary time direction and the environment region $\overline{A}$ for each replica is independent in the imaginary time direction. Therefore, the imaginary time length for $\mathcal{H}_A$ is $\beta_A = n$ and that for total system $\mathcal{H}$ is $\beta = 1/T$.

## Results

### The scheme to extract the entanglement spectra

The ES of a subsystem $A$ coupled with environment $\overline{A}$ is constructed via the reduced density matrix (RDM), defined as the partial trace of the total density matrix $\rho$ over a complete basis of $\overline{A}$, $\rho_A = \mathrm{Tr}_{\overline{A}}\rho$. The RDM $\rho_A$ can be interpreted as an effective thermodynamic density matrix $e^{-\mathcal{H}_A}$ through an entanglement Hamiltonian $\mathcal{H}_A$. As known in closed system, the spectral function $S(\omega)$ for physical observable, represented as $\mathcal{O}$, can be written by the eigenstates $|n\rangle$ with the eigenvalue $E_n$ of the Hamiltonian $\mathcal{H}$,

$$S(\omega) = \frac{1}{\pi}\sum_{m,n} e^{-\beta E_n} |\langle m|\mathcal{O}|n\rangle|^2 \delta(\omega - [E_m - E_n]). \quad (1)$$

Therefore, there is a relation between energy spectrum $S(\omega)$ and imaginary time correlation $G(\tau)$ as $G(\tau) = \int_0^\infty d\omega K(\omega, \tau)S(\omega)$. The $K(\omega, \tau)$ is a kernel with slightly different expressions for bosonic/fermionic $\mathcal{O}$[66–76]. The energy spectrum of the corresponding operator can be analytically continued from the correlation function in imaginary time. However, the relation between RDM and the modular Hamiltonian[45–48], $\rho_A = e^{-\mathcal{H}_A}$, doesn't contain any information of an effective imaginary time $\beta_A$ of the subsystem. To compute ES, the first task is to construct a "partition function" of $\mathcal{H}_A$ with effective imaginary time $\beta_A$.

The solution comes from the n-th order of RDM, $\rho_A^n$, which can be written as $\rho_A^n = e^{-n\mathcal{H}_A}$. In this way, we can readily make use of such effective imaginary time $\beta_A = n$ at $n = 1, 2, 3, \cdots$ integer points. It's similar to how the Rényi EE is computed in QMC via the replica partition function[19,21,55,61,64,78,79],

$$\mathcal{Z}_A^{(n)} = \mathrm{Tr}[\rho_A^n] = \mathrm{Tr}[e^{-n\mathcal{H}_A}]. \quad (2)$$

As depicted in Fig. 1, $\mathcal{Z}_A^{(n)}$ is a partition function in a replicated manifold, where the boundaries of area $A$ of the $n$ replicas are connected in imaginary time and the boundaries of the area $\overline{A}$ are independent (for sites in $\overline{A}$ for each replica, the usual periodic boundary condition of $\beta$ is maintained). It can be seen that the effective $\beta_A = n$ of the subsystem $A$ is in the unit of integer numbers whereas the $\beta = 1/T$ of the total system is in the inverse unit of the physical energy scale of the original system, $J$ of the Heisenberg model, for instance. We note a similar approach was carried out in the interacting fermion systems[37,38,44] via determinant quantum Monte Carlo (DQMC), although the entangling region therein is still (quasi) 1D. However, we think the relative complicated mathematical derivation in the DQMC to translate an interacting fermionic partition function into sampling of determinants and the lack of the simple but deep wormhole picture in the previous works, may obscure the widespread usage of the computation of ES in QMC, for example, we were not aware of these works as we carried our independent wormhole approach. In this regard, our work is the first one in doing so in boson/spin QMC, and our path-integral interpretation and the wormhole picture greatly simplified the key physical idea of the computation of ES in QMC simulations with the first computation of ES for a 2D entangling region presented. We hope our presentation will make such computation more accessible to the broader audience.

As shown in Fig. 1, since the imaginary times $\tau_A$ between two nearest replicas differs by $\delta\tau_A = 1$ (an effective imaginary time evolution operator $e^{-\mathcal{H}_A}$ of subsystem acts between them), we can measure the imaginary time correlation function at integer points, to obtain $G(\tau_A)$ of $\tau_A = 0, 1, 2, \ldots, n$. The correlation function $G(\tau_A) \sim e^{-\Delta\tau_A}$ when $\beta_A \to \infty$ where $\Delta$ is the lowest energy gap in the ES. When there is large gap, the $G(\tau_A)$ decays very fast and it is very hard to extract the high energy part of spectrum because the distance $\delta\tau_A = 1$ is not small enough. However, since the important information of ES is usually encoded in the low-lying spectra, it can be obtained with controlled accuracy from the long-time imaginary time correlations in QMC simulations, i.e. the

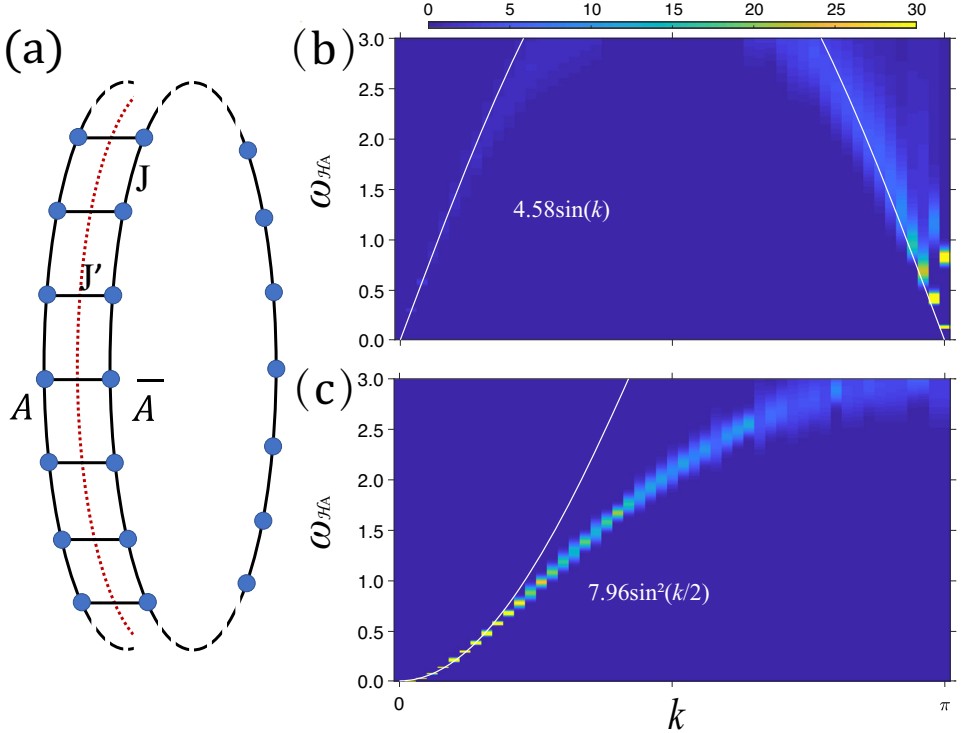

**Fig. 2 | ES of Heisenberg ladder. a** Heisenberg spin ladder. The red dashed line cut it into two entangled constituents, $A$ and $\overline{A}$. **b** The low-lying ES with $L = 100$, $J' = 1.732, J = 1$ and $\beta = 100$, $\beta_A = 200$. The white line is fitting to the data with the dispersion $4.58\sin(k)$. **c** The low-lying ES with $L = 100$, $J' = 1.732, J = -1$ and $\beta = 100$, $\beta_A = 800$. The white line is fitting to the data with the dispersion $7.96\sin^2(k/2)$.

more number of replicas $n$, the longer the imaginary time correlation $\tau_A$, and lower "energy" $\omega_{\mathcal{H}_A}$ in ES accessed. With the good quality $G(\tau_A)$ at hand, the stochastic analytic continuation (SAC) scheme can reveal reliable spectral information, $S(\omega_{\mathcal{H}_A}(k))$, as have been widely tested in fermionic and bosonic quantum many-body systems in 1D, 2D and 3D[66–68,70–74,80–86]. In the following two examples, we use stochastic series expansion (SSE) QMC for quantum spin systems[87–91] combined with SAC to obtain the related ESs. All the imaginary time correlations are computed via spin $S^z$ operators, i.e., $G_k(\tau_A) = \langle S^z_{-k}(\tau_A) S^z_k(0) \rangle$.

## Spin-1/2 Heisenberg ladder

As the first example to demonstrate the power of our method, we compute the ES of the Heisenberg ladder with $L = 100$ and compare with the ED results in small sizes $L = 10, 12, 14$[27]. The spins on the ladder are coupled through nearest neighbor Heisenberg interactions as shown Fig. 2a, with the strength $J$ along the leg and $J'$ on the rung. The Hamiltonian of the spin-1/2 ladder is

$$H = J \sum_{\langle ij \rangle} (S_{A,i} S_{A,j} + S_{\overline{A},i} S_{\overline{A},j}) + J' \sum_i S_{A,i} S_{\overline{A},i} \qquad (3)$$

Here, $S_{A,i}$ and $S_{\overline{A},i}$ are spin-1/2 operators at site i of $A$ and $\overline{A}$. $\langle i,j \rangle$ denotes a pair of nearest-neighbor sites on the spin chain of $L$ sites with periodic boundary conditions. $J$ is the coupling strength intra-chain and $J'$ is coupling inter-chains.

We first simulate with $J = 1$ and $J' = 1.732$ at $\beta = 100$ and $\beta_A = 200$ (200 replicas). Our QMC ES in Fig. 2b are consistent with the ED results, but our larger system sizes clearly reveal new features at the thermo-dynamic limit (TDL). First, with $L = 100$, the finite size gaps at $k = 0$ and $\pi$ are much smaller than the ED results with $L = 14$, e.g. $\Delta(\pi) \sim 0.48$ in ED [Fig. 3a in ref. 27] whereas $\Delta(\pi) \sim 0.1$ in QMC, suggesting the gap closes at TDL. Second, the ES here is expected to bear the low-energy CFT structure, i.e., the ground level of ES, $\xi_0$, will scale as $\xi_0/L = e_0 + d_1/L^2 + \mathcal{O}(1/L^3)$ where the $d_1 = \pi c v / 6$ according to the CFT

predication with the central charge $c = 1$ and $v$ the velocity of ES from the entanglement Hamiltonian near its gapless point[27], i.e. the Cloizeaux-Pearson spectrum of the quantum spin chain, $v|\sin(k)|$[92]. The ED fit at $L = 14$ gave $v \sim 2.36$ [from the Fig. 3a in ref. 27], however, the fitting of QMC reveals the $v \sim 4.58$, as shown by the fitting line in Fig. 2b. This again reflects ES is greatly affected by finite size effect and it is necessary to access larger system sizes for quantitatively correct information.

Furthermore, we simulate the cases with both ferromagnetic (FM) $J = -1$ and antiferromagnetic (AFM) $J' = 1.732$ at $\beta = 100$ and $\beta_A = 800$ on the same ladder to compare with the results in ref. 27, where the ladder is in the gapped rung singlet phase. Since the ES in TDL is expected to show the spectrum of FM spin chain, i.e. quadratic dispersion $\sim \sin^2(k/2)$ close to $k = 0$[93,94], we purposely chose $\beta_A = 800$ such that more low-lying ES can be obtained. As shown in Fig. 2c, the obtained dispersion of ES indeed resembles that of an edge Hamiltonian of the FM chain. In the Fig. 4e of ref. 27, a linear dispersion is found and it was attributed to the finite size effect. Now that we can access $L = 100$, the ES indeed reveals a quadratic dispersion $7.96\sin^2(k/2)$ as shown by the white line therein. The results in Fig. 2b, c, clearly demonstrate the correspondence between the ES and the true spectra of the edge Hamiltonian, consistent with the previous ED study, and achieve the TDL readily.

## Equivalence principle of ES

We use the equivalence principle to understand why the ES resembles the energy spectrum of the subsystem within the frame of the replicas. The schematic diagram of the path integral of the RDM is shown in Fig. 3a, the x/y/z axis represents the direction of the $J'$ bond/$J$ bond/ imaginary time, respectively. Because both the subsystem $A$ and environment $\overline{A}$ are spin chains, there are only two pieces paper-like configurations along x-axis in the figure, that is, whole of the chain is an entangled edge without bulk. The wavy lines mean the interactions between subsystem $A$ and environment $\overline{A}$. The $\overline{A}$ has been divided into

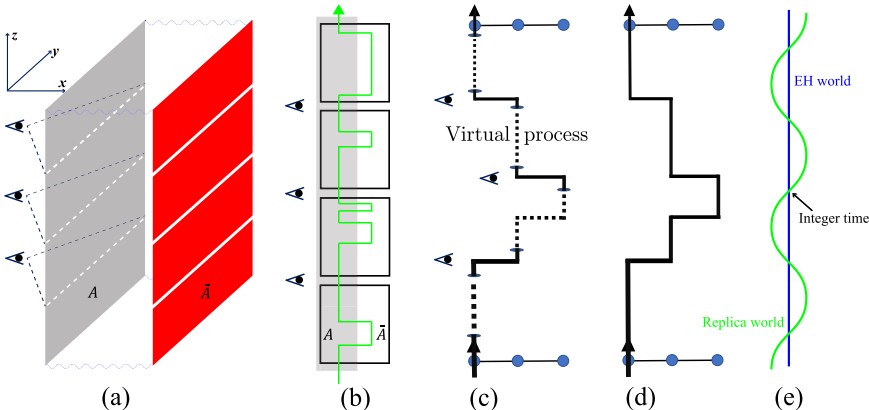

**Fig. 3 | The equivalence principle of ES and energy spectrum. a** The schematic diagram of the path integral of the reduced density matrix. The x/y/z axis represents the direction of the $J'$ bond/$J$ bond/imaginary time. The wavy lines mean the interactions between subsystem $A$ and environment $\overline{A}$. The $\overline{A}$ has been divided into several disconnected replicas along the imaginary time axis while all the replicas of $A$ are connected in time, which is the same as the Fig. 1. The eyes observe the information of ES at the integer time point of $\beta_A$ which means we can only extract the information of ES on the connections of replicas. **b** The $x - z$ plane view of (**a**). The gray part is the subsystem $A$ and the white part is the environment $\overline{A}$. The black plaquettes mean the replicas and the green line represents the worldline of a particle (spin). **c** A worldline evolution in a replica system under the view of $y - z$ plane in (**a**). The real worldline (solid line) locates at the $\tau_A = 0, 1, 2, \ldots$ layers and the line goes in/out of a hole means it goes in/out of one replica. Between a pair of holes, we can imagine a virtual process (dashed line) which is equivalent to a real process in normal system. **d** A worldline evolution in a normal path-integral of closed system. **e** The blue line means the space-time world of entanglement Hamiltonian (EH) which is time independent. The green line represents the space-time world of replica manifold, i.e., the system drawn in (**a**). These two worlds cross at integer time points, thus, we can obtain the information of EH from the green system at integer time.

several disconnected replicas along the imaginary time axis while all the replicas of $A$ are connected in time. It's worthy noting that the evolution inside one replica can not reflect the information of ES. As mentioned above, we have to measure the ES information at the connections of replicas. As drawn in the Fig. 3a, the eyes observe the information of ES at the integer time point of $\beta_A$.

The Fig. 3b is the $x - z$ plane view of Fig. 3a. The gray part is the subsystem $A$ in which all the replicas are connected and the white part is the environment $\overline{A}$ in which all the replica are disconnected. At the same time, the black plaquettes mean the replicas and the green line represents the worldline of a particle (spin). From this figure, we note that even though the worldline can walk around everywhere inside a replica, it has to go back to $A$ region on the connections where we observe the information of ES. Thus, what we see from the measurement is an effective system as large as $A$, even though we simulate both $A$ and $\overline{A}$. So far, it just explains the spatial size of the entanglement Hamiltonian is the same as the subsystem $A$, but we haven't explained why the entanglement Hamiltonian resembles the Hamiltonian of $A$.

With the above picture, it is easy to understand the working of Haldane's conjecture in the path integral, i.e. the entanglement spectrum of $A$ is similar with the energy spectra of a closed system $A$ without coupling to $\overline{A}$. Fig. 3c is that of a replicated system (Heisenberg ladder) under the view of $y - z$ plane in Fig. 3a, d is an example of the evolution of worldline in such a closed system (Heisenberg chain). For the replicated system, we can measure the physical observables at the integer $\beta_A$ points to extract the information of ES, denoted by the observing eyes, and the physical rules (e.g., Hamiltonian operators and worldlines) one sees at these time points of replica manifold should be similar as in a closed system $A$ controlled by at the same Hamiltonian operator imaginary times. In short, what is observed in the replica system is the Heisenberg Hamiltonian operators and the corresponding worldline evolution, so ES is very much like a Heisenberg model of subsystem $A$.

The reader may ask that even if the rules on the connections of replicas (integer imaginary time) resemble the closed system, how about the rules of other imaginary times? As shown in Fig. 3c, the real worldline (solid line) means it is at the $\tau_A = 0, 1, 2, \ldots$ we can measure, i.e., the connected part of two nearest replicas, and the virtual process goes in/out of one replica are denoted as the dashed line. Furthermore,

because $\mathcal{H}_A$ is independent on imaginary time, the rule of worldline evolution should be same in any time, i.e., the translation invariance of time. Thus, $\mathcal{H}_A$ is likely to be the similar as $\mathcal{H}$ of real closed $A$ system all the time. A more intuitive diagram can be seen in Fig. 3e, the blue line means the space-time world of EH which is time independent. The green line represents the space-time world of replica manifold, i.e., the system drawn in (a). These two worlds cross at integer time points, thus, we can obtain the information of EH from the green system at integer time.

The above argument can be applied to the case where every sites of subsystem $A$ is coupled with environment $\overline{A}$, i.e., the $A$ has only entangled edge without bulk. It should not only work in the ladder case, but should also work in other similar cases, such as bilayer. To further verify our argument of the equivalence principle, we simulate a coupled bilayer system, in which one layer is the subsystem $A$ and the other is the environment $\overline{A}$. We find the ES is indeed similar to the energy Hamiltonian of a closed one layer system, even the system goes through a quantum phase transition. The results are shown below.

**Antiferromagnetic Heisenberg Bilayer**

An AFM Heisenberg model on bilayer square lattice is shown Fig. 4a, where $J$ and $J'$ are the intra- and inter-layer couplings. We define the spin-1/2 Hamiltonian on a bilayer lattice via the same equation as Eq. (3), where $S_{A,i}$ and $S_{\overline{A},i}$ are spin-1/2 operators at site i of $A$ and $\overline{A}$, $\langle i,j \rangle$ denotes a pair of nearest-neighbor sites on the square lattice of $L \times L$ sites with periodic boundary conditions.

We compute the ES with the bottom layer as $A$ and the top layer as $\overline{A}$. The (2 + 1)d O(3) QCP, separating the Néel phase and inter-layer dimer product state, is found to locate at $J'/J = 2.5220(1)$ from high-precision QMC simulations[19,22,95,96]. The EE of antiferromagnetic Heisenberg bilayer has been studied in ref. 97, but the ES is still lacking. We simulate three cases to demonstrate our prediction: $J'/J = 1.732$ in the Néel phase [Fig. 4b], $J'/J = 2.522$ at the critical point [Fig. 4c], $J'/J = 3$ in the dimerized phase [Fig. 4d] at $\beta = 100$ and $\beta_A = 32$ with size $L = 50$.

All the three cases strongly support our understanding: the ESs have two gapless modes with a strong one at $(\pi, \pi)$ and a weak one at $(0, 0)$, closely resembling those the Goldstone modes in square

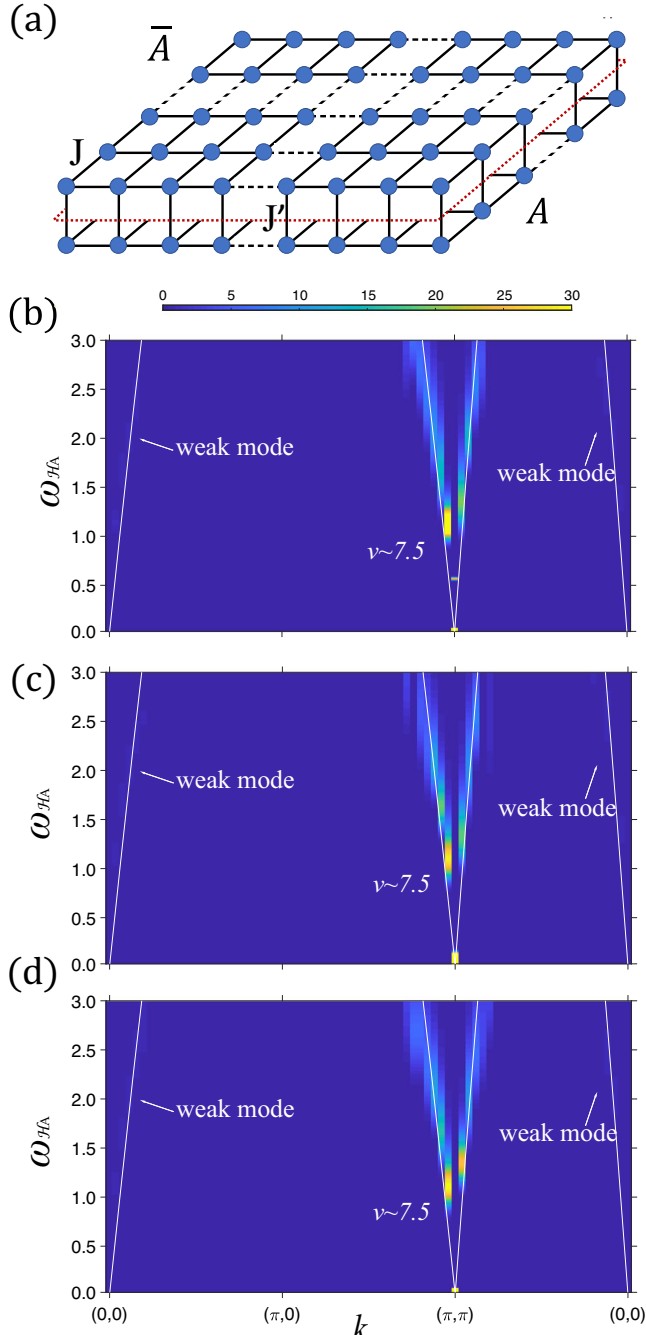

**Fig. 4 | ES of Heisenberg bilayer across (2+1) O(3) transition. a** Antiferromagnetic Heisenberg bilayer. The red dashed line cut it into two entangled constituent layers, $A$ and $\overline{A}$. **b** The low-lying ES of in the Néel phase with $L = 50$, $J' = 1.732$, $J = 1$ and $\beta = 100$, $\beta_A = 32$. **c** The low-lying ES at the quantum critical point with $L = 50$, $J' = 2.522$, $J = 1$ and $\beta = 100$, $\beta_A = 32$. **d** The low-lying ES in the dimerized phase with $L = 50$, $J' = 3$, $J = 1$ and $\beta = 100$, $\beta_A = 32$. We show the obtained ES along the high-symmetry path $(0,0) - (\pi, 0) - (\pi, \pi) - (0, 0)$. The white line is fitting to the data with the dispersion $7.5\sqrt{1 - \frac{1}{4}(\cos(k_x) + \cos(k_y))^2}$, the linear spin wave for anti-ferromagnetic Heisenberg model on square lattice.

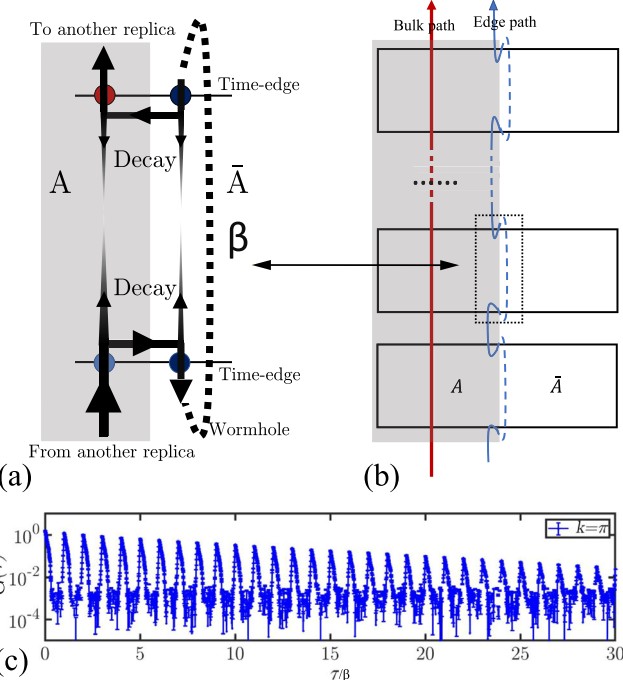

**Fig. 5 | The wormhole effect. a** Wormhole effect of worldlines going through a replica. It is a zoom-in of the region inside dotted box of (**b**). The gray part is the subsystem $A$ and the white part is the environment $\overline{A}$. The arrows go into bulk will decay to zero as $\beta \to \infty$. At the same time, the arrows go through the imaginary-time-edge of environment will reach the other side through "wormhole" without much attenuation. **b** The path integral of replica system which has both bulk and edge. The gray part is the subsystem $A$ and the white part is the environment $\overline{A}$. The red/blue line represents the worldline path inside the bulk/on the edge. **c** The $G_{k=\pi}(\tau)$ along imaginary time direction of the replica system for AFM spin ladder in Fig. 2b with $L = \beta = 100$ and $\beta_A = 100$. The fast mode inside one replica and the slow mode between the replicas are clearly seen. **c** A worldline evolution in a normal path-integral of closed system.

## Wormholes in the path integral

Although the equivalence principle (Fig. 3) well explains the relation between the ES and energy spectra in above cases, both these systems have only entangled edges without bulk, they are special. We find the geometrical manifold of replicas (Fig. 1) provides a very intuitive picture to understand the general Haldane's conjecture. It points to a wormhole effect in the space-time of the replica manifold to make the edge mode of energy spectrum more important in the low-lying ES. As shown in Fig. 5a, b, the gray part is the subsystem $A$ and the white part is the environment $\overline{A}$ and (a) is a zoom-in of the region inside dotted box of (b). It's worthy noting that the replica system is much thicker here than in Fig. 3b, which represents it has both edge and bulk. As shown in Fig. 5a, when a worldline (black line) goes into one replica, it has many choices for the possible paths. The imaginary time correlations caused by the worldline will decay to zero if the worldline goes straight into depth of replica along time direction, because the correlation function $G(\tau) \sim e^{-\Delta \tau}$ and the time length inside replica is infinite when $\beta \to \infty$. At the same time, the other worldline goes through the imaginary-time-edge of environment $\overline{A}$ will reach the other side through "wormhole" without much attenuation. Because tracing $\overline{A}$ actually provides a wormhole-like escapeway through connecting both imaginary-time-edges of environment, i.e., the periodic boundary condition (PBC) of the imaginary time of $\overline{A}$. It automatically guarantees the imaginary time correlations near the connection between replicas is stronger, which contribute to the ES. This picture is proved in our numerics via the correlation function along imaginary time direction in Fig. 5c, where the $G_{k=\pi}(\tau)$ for AFM Heisenberg ladder [the case of

antiferromagnetic Heisenberg model[70,72,98]. We therefore fit all the three ESs in Fig. 4b, c and d with the smilar linear spin wave[99,100] dispersions. As far as we are aware of, these results serve as the first measurement of ES in 2D entangling region, also consistent with the equivalence principle of ES and our wormhole picture of worldline in QMC simulation of ES, as we now turn to.

Fig. 2b] is shown. It is obvious that there are two time/energy scales: The fast-decaying one is led by the original Hamiltonian, which happens inside replica between two integer $\beta_A$ points. It goes into the depth of replica along time direction as the "decay" ones in Fig. 5a. The slow-decaying one (envelope of upper boundary at the $\beta_A = 0, 1, 2, \ldots$) is generated by the wormhole effect, which reflects the imaginary time dynamics of the entanglement Hamiltonian. It obviously demonstrated that the wormhole effect strengths the imaginary time correlations of EH, that is, leads the low-lying ES.

We note such wormhole effect in the path-integral formulation is for the first time reported, and it not only offers the explanation of the Li and Haldane conjecture, but also predict new phenomena that can be tested within the replicated manifold settings[77], as we discuss below.

### Amplification of bulk energy gap in ES

Our wormhole picture of the path-integral in replicated manifold for ES is very general. Although the ladder and bilayer we studied above belong to the subsystem $A$ has only edge without bulk, the wormhole effect also plays important role in the systems with both bulk and edge. In the following, we would discuss Handane's conjecture in these cases.

As the summation of weights in path integral is $e^{-\int \mathcal{D}(\mathcal{L})\Delta(\mathcal{L})}$, where the $\mathcal{L}$ is the path and $\Delta(\mathcal{L})$ is the gap of this path. In a rough mean field estimation, the cost can be treated as $\overline{\mathcal{L}} \times \overline{\Delta(\mathcal{L})}$. The $\overline{\mathcal{L}}$ is the (imaginary time) path length and $\overline{\Delta(\mathcal{L})}$ is the mean gap along this path. The smaller cost $\overline{\mathcal{L}} \times \overline{\Delta(\mathcal{L})}$ make the related weight of the path integral more important.

Comparing the two typical paths in the depth of bulk and around the entangled edge as the red line and blue line shown in the Fig. 5b, it's obvious that the edge mode takes more important role. Under a rough estimation, the scale of the red path length is about $\beta \times n$ and the blue path length is about $1 \times n$. The $n$ is the number of the replicas. Thus, the ratio of the two paths can be simplified as $\beta : 1$. Therefore, wormhole effect gives the ratio of the cost between the bulk and entangled edge as $\beta \overline{\Delta_b} : \overline{\Delta_e}$. The energy gap of bulk is amplified by a factor of $\beta$ in the ES and it makes the low-lying ES always close to the edge energy mode at $\beta \to \infty$ limit, not only in the topological state cases (gapped bulk and gapless edge), but also both bulk and edge have finite gaps. We therefore conject that if the $\beta$ is finite, the entangled edge are gapped and subsystem bulk is gapless, the low-lying ES will be more like the energy spectra of bulk. Such "reversal" of the Li and Haldane conjecture has been shown in recent work[77]. Moreover, if both bulk and edge are gapped, the $\beta$ will lead to a competition of both gaps and induce a transition of the ES at finite temperature. We note the similar dynamical behaviors (amplification of bulk gap) has also been observed in 1D system when it can be described by CFT[101]. However, our result is beyond CFT and dimension, and is more general and fundamental. We also note that the forms of the entanglement (modular) Hamiltonian across such transitions have been discussed in the rich literature[45–48], it is possible to foresee if the entanglement (modular) Hamiltonian exhibits a local structure, then one could in principle extract (or "learn") the entanglement Hamiltonian from the time evolution of local observables in QMC. We leave these interesting directions to future works.

## Discussion and conclusion

Overall, we realize a practical scheme to extract the low-lying ES from QMC simulations and the computation of ES for 2D entangling region is presented for the first time. Combined with the unifying picture of equivalence principle and wormhole effect in path-integral formulation, our method makes the ES measurement possible for high dimension quantum many-body systems with large entangling region. Our method is not only limited to QMC for spins, as the existed pioneering works in computing the ES for interacting fermion systems[37,38,44], but can also be extended to other numerical approaches

for highly entangled quantum matter, such as the finite temperature tensor-network algorithm[102–104].

## Methods

### Stochastic analytical continuation

We employ a stochastic analytical continuation (SAC)[66–69,75] method to obtain the spectral function $S(\omega)$ from the imaginary time correlation $G(\tau)$ measured from QMC, which is generally believed a numerically unstable problem.

The spectral function $S(\omega)$ is connected to the imaginary time Green's function $G(\tau)$ through:

$$G(\tau) = \int_{-\infty}^{\infty} d\omega S(\omega) K(\tau, \omega) \tag{4}$$

here $K(\tau, \omega)$ is the kernel function depending on the temperature and the statistics of the particles. We restrict ourselves to the case of spin systems and with only positive frequencies in the spectral, where $K(\tau, \omega) = (e^{-\tau\omega} + e^{-(\beta-\tau)\omega})/\pi$. Then, we have

$$G(\tau) = \int_0^{\infty} \frac{d\omega}{\pi} \frac{e^{-\tau\omega} + e^{-(\beta-\tau)\omega}}{1 + e^{-\beta\omega}} B(\omega) \tag{5}$$

here $B(\omega) = S(\omega)(1 + e^{-\beta\omega})$ is the renormalized spectral function.

In fact, $G(\tau)$ for a set of imaginary time $\tau_i$ is obtained by QMC with the statistical errors. The renormalized spectral function can be set into large number of equal-amplitude $\delta$-functions

$$B(\omega) = \sum_{i=0}^{N_\omega} a_i \delta(\omega - \omega_i) \tag{6}$$

Then the fitted Green's functions $\tilde{G}_i$ from Eq. (5) and the measured Greens functions $\bar{G}_i$ are compared by the fitting goodness

$$\chi^2 = \sum_{i,j=1}^{N_\tau} (\tilde{G}_i - \bar{G}_i)(C^{-1})_{ij}(\tilde{G}_j - \bar{G}_j) \tag{7}$$

where the covariance matrix is defined as

$$C_{ij} = \frac{1}{N_B(N_B - 1)} \sum_{b=1}^{N_B} (G_i^b - \bar{G}_i)(G_j^b - \bar{G}_j), \tag{8}$$

with $N_B$ the number of bins, the measured Green's functions of each $G_i^b$.

The weight for a given spectrum is taken to follow a Boltzmann distribution via Metropolis sampling

$$W(\{a_i, \omega_i\}) \sim \exp\left(-\frac{\chi^2}{2\Theta}\right) \tag{9}$$

with $\Theta$ a virtue temperature to balance the goodness of fitting $\chi^2$ and the smoothness of the spectral function. All the spectral functions of sampled series will be averaged to obtain the final spectrum.

In this paper, we use the correlation of $S^z$ operators to obtain the spectra. For a broad class of spin models, the operator $S^z$ can reveal a lot excitations unless it can not connect the ground state and the excited state (e.g., the excitation changes the total $S^z$). For example, previous works on extracting the spectra of other models even without SU(2) symmetry via $S^z$ correlations: neutron scattering spectrum through transverse field Ising model[105]; interaction between visons in quantum dimer model[71]; and the Higgs mode in a weak Zeeman field Heisenberg model[72], etc.

In addition, we will need to design different operators to probe some special spectra. These can be readily implement in many other systems, for instance, people have measured the vison and dimer

correlations and their spectra in quantum dimer models[71,74,76], the $S^+S^-$ and bond-bond correlations and their spectra in quantum spin models[70,72-74]. The corresponding entanglement spectral measurements, can be carried out in the similar manner.

## Data availability

The data that support the findings of this study are available from the authors upon reasonable request.

## Code availability

All numerical codes in this paper are available upon reasonable request to the authors.

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

## Acknowledgements

We would like to thank Bin-Bin Chen, Meng Cheng, Hui Shao, Yan-Cheng Wang, Jiarui Zhao and Bin-Bin Mao for fruitful discussions. We also thank Fabien Alet and Fakher Assaad for useful comments. We acknowledge support from the Research Grants Council of Hong Kong SAR of China (Grant Nos. 17303019, 17301420, 17301721, AoE/P-701/20 and 17309822), the K. C. Wong Education Foundation (Grant No. GJTD-2020-01) and the Seed Funding "Quantum-Inspired explainable-AI" at the HKU-TCL Joint Research Centre for Artificial Intelligence. Z.Y. thanks the start-up fund of Westlake University. We thank the HPC2021 platform under the Information Technology Services at the University of Hong Kong and the Tianhe platforms at the National Supercomputer Centers in Tianjin and Guangzhou for their technical support and generous allocation of CPU time. The authors also acknowledge Beijng PARATERA Tech Co.,Ltd.(https://www.paratera.com/) for providing HPC resources that have contributed to the research results reported within this paper.

## Author contributions

Z.Y. developed the QMC algorithm and put forward the wormhole mechanism. Both authors contributed to the analysis of the results. Z.Y.M. supervised the project.

## Competing interests

The authors declare no competing interests.
