## [Peer Review File · Nature Communications]

REVIEWER COMMENTS

Reviewer #1 (Remarks to the Author):

This work proposed a new approach to study low (entanglement) energy entanglement spectrum using quantum Monte Carlo method. The key insight is that entanglement spectrum can be obtained from spectral functions, when the reduced density operator is treated as the imaginary time evolution operator. This method is applied to spin models in one and two spatial dimensions. The authors provided a physical interpretation of nontrivial entanglement spectrum when two gapless systems are coupled to obtain a gapped state, which they called the wormhole effect. I think the results of this paper are quite novel and interesting, and thus would like to recommend it for publication in Nature Communication, after the following comments are addressed.

1. I found that the discussion of wormhole effect is not so clear. For example the authors used the term of "bulk" and "boundary" and it is not clear what they refer to, since the entanglement cut considered is between two layers, rather than across the boundary of a spatial region. I found it difficult to understand Fig 3 and the text related to it. What does the width of the arrows mean in Fig 3?

2. Overall there are too many English errors in the draft which make it difficult for the readers. For example, in Page 4, "Furthermore, because HA is independent

on imaginary time, the rule of worldline evolution...hence the Li and Haldane's conjecture". In these three sentences there are many grammar errors and I found it difficult to guess its meaning.

3. In Page 5, the authors discuss that there is a phase transition at finite replica index n , and suggest that there must be long-range interaction in entanglement Hamiltonian. I am wondering if it is possible to directly verify the long range order by computing two-point functions for two far-away points, rather than relying on the existence of Goldstone modes.

Reviewer #2 (Remarks to the Author):

This manuscript discusses a scheme for extracting the low-lying entanglement spectrum from world-line Quantum Monte Carlo (QMC) simulations and draws connections between the Li-Haldane

conjecture and the wormhole effect in the path integral formulation of the reduced density matrix. Extracting information about the entanglement spectrum is highly relevant in quantum many-body physics but at the same time a very challenging task. The importance of this topic stems from the fact that the low-lying entanglement spectrum contains information about the topology of the system, i.e. according to the Li-Haldane conjecture, there is a one-to-one correspondence between the entanglement spectrum and the spectrum of edge excitations of the system. While the calculation of the low-lying entanglement spectrum is usually performed via wave-function-based methods, these methods show good performance for 1 dimensional systems but are difficult to apply in higher dimensions. Therefore, with the calculation of the entanglement spectrum from QMC methods and for higher-dimensional systems, this paper addresses a very relevant and timely topic.

I think the paper, even though well-written, is pretty difficult to understand. I would encourage the authors to make an effort to explain some of the basic concepts better and back up the explanation with graphical illustrations. Part of that could be done by expanding the supplemental material, which is very short. Additionally, the authors should try to explain a bit better the novelty of their method with respect to other proposals for extracting the entanglement spectrum in QMC simulations. Is this really the first time this has been done? If these points are addressed convincingly, I would be supportive of publication in Nature Communications. I will give more details below.

The paper can very clearly be separated into 2 parts: The first part discusses the method for obtaining the low-lying entanglement spectrum from imaginary time correlation functions and the replica partition function of the reduced density matrix. The second part tries to explain the Li-Haldane conjecture via a wormhole picture in the worldline evolution of the replica system.

The first part sounds to me like a good and pretty obvious idea. Since the spectrum in MC is evaluated from correlation functions in imaginary time, it sounds quite natural to me to introduce many copies of the system such as to mimic imaginary time evolution with the modular Hamiltonian H_A at integer times. One actually wonders why this idea was not pursued earlier in the QMC community. I have to admit that I do not really understand the difference (the novelty) with respect to section III in the paper [\href{https://journals.aps.org/prb/abstract/10.1103/PhysRevB.89.125121}](https://journals.aps.org/prb/abstract/10.1103/PhysRevB.89.125121){Phys. Rev. B **89**, 125121}. The authors should try to explain better how their method is related to that as well as to other approaches of extracting the entanglement spectrum in QMC. Other than that, the results look convincing. I would suggest adding to Figure 4 c) the exact results for $L = 14$ which show a linear dispersion and contrast it with the quadratic dispersion which is revealed from the QMC simulations on larger system sizes.

I have to admit that (even after reading the section several times) I do not completely understand the wormhole picture the authors try to convey. I would suggest the authors make an effort trying to explain this better, maybe with a dedicated section in the supplemental material. Figure 3 (a), (c), and (d) are confusing to me. It is not entirely clear what's on the axis and maybe one should really expand this picture in the Supplemental Material instead of just writing "from another replica" and "to another replica".

If I understand correctly, from similarities (equivalence) of the world-line evolutions of the closed system and the replica system, the authors conclude that \mathcal{H} is similar to \mathcal{H}_A . I would like to make the authors aware that the modular Hamiltonian for ground states has been investigated in several works (see e.g. <https://www.nature.com/articles/s41567-018-0151-7> {Nature Physics volume **14**, pages 827-831 (2018)}, <https://journals.aps.org/prb/abstract/10.1103/PhysRevB.98.134403> {Phys. Rev. B **98**, 134403} or <https://journals.aps.org/prb/abstract/10.1103/PhysRevB.100.155122> {Phys. Rev. B **100**, 155122}). Here it is shown that the modular Hamiltonian has a similar "local" structure as the system Hamiltonian, but different coupling coefficients which in a way define a local "entanglement temperature". Would it be possible to confirm this picture in QMC simulations? Would it make sense to compare the world-line evolution of such a "deformed" modular Hamiltonian with the one of the system Hamiltonian?

Related to the above: In recent work, it has been shown that parent Hamiltonians of Gibbs states can be reconstructed from local observables on the system (see e.g. <https://quantum-journal.org/papers/q-2019-07-08-159/> {Quantum **3**, 159 (2019)}). Given the above that the modular Hamiltonian exhibits a local structure, these methods could be applied to extract the entanglement Hamiltonian from local observables measured in QMC simulations. Would it be conceivable to extract the entanglement spectrum from imaginary time evolution with the "learned" modular Hamiltonian? What do the authors think of how such a method would compare to their proposal? Would it make sense to add a comment on this in the paper?

Reviewer #3 (Remarks to the Author):

The authors present a study of the entanglement spectrum of quantum spin models via large-scale quantum Monte Carlo simulations in one and two dimensions. The results are significant, since the system sizes studied are an order of magnitude larger than previous studies, and the first computation of the entanglement spectrum for a 2D entangling region is presented. The results

support the conjecture by Haldane that the entanglement spectrum has a close correspondence with the energy spectrum of the subsystem in isolation.

The results have been obtained by a novel combination of existing cutting edge techniques, namely the replica time-displaced correlation function introduced in ref [41], and the stochastic analytic continuation method. The resulting data is scientifically sound with a high degree of confidence.

A main focus of the paper is on the so-called “wormhole effect,” whereby correlations decay exponentially in imaginary time, but experience revivals due to the presence of replicas. Though this is an interesting effect which is being presented for the first time, is not clear that this observation impacts the results or conclusions of the paper in any way. Indeed ref [41] studied the replica time-displaced correlation function only at integer time displacements (only at the revival peaks), representing integer powers of the reduced density matrix. As such I don’t believe this effect should be a centerpiece of this paper, and I think naming it the wormhole effect is potentially misleading.

Overall, I think that this is a significant numerical study of entanglement spectra in large-scale quantum Monte Carlo simulations, which would be prohibitive with other methods. However, I find that the impressive results have been overshadowed by an unnecessary emphasis on a curious, but not surprising, effect with a sensationalized name (the wormhole effect). I would recommend publication in nature communications provided that the emphasis in the paper be significantly redirected to focus on the results and the novel implementation of the methods. Below I will give a list of suggested improvements.

1) I think the title is too sensational, and I haven’t been convinced that the wormhole effect explains Haldane’s conjecture.

2) For a balanced view, I think it is important to mention references that are critical of the supposed universality of the entanglement spectrum, for instance Phys. Rev. Lett. 113, 060501.

3) In the introduction, general statements about the Monte Carlo method are made, but the supporting citations are specific studies by one of the authors (ref [49] and [50]), which don’t seem to fit.

4) From the introduction: “Here we first develop a protocol to overcome the exponential growth of computational complexity in obtaining the ES via QMC combined with stochastic analytic continuation.” As far as I know, this technique has already existed in ref [41] and it was pointed out

in ref [42] that it could be used for stochastic series expansion Monte Carlo for spin systems as done in this work.

5) It is frequently stated that there are comparisons with exact diagonalization (ED) from reference [27], although this data is not included in the main text and is not clear how to compare with the given reference. This data should be presented in the main plot. Moreover, I think it is very important to give direct comparisons with exact results on small system sizes. Can't one use the QMC technique on $L=10$ and compare with exact results directly?

6) Many branches of the spectra cannot be seen by eye without zooming in as much as possible on a pdf. Perhaps changing the color scale would make these stand out?

7) Velocities are extracted, do these agree with theoretical predictions? This fitting for ED is not shown.

8) From wormholes in the path integral section: "one can use a wormhole effect of the worldline in QMC simulation in space-time to explain the tracing of the environment in the path- integral." The logic here is reversed. The tracing of the environment explains the the wormhole effect.

9) I find figure 3c and 3d a bit confusing. I think the point that worldlines can make shortcuts through imaginary time by travelling through replicas is well explained already from fig 3a.

10) I find the second paragraph of the "wormholes in the path integrals" section particularly confusing. Presumably this is where the connection between the wormhole effect and Haldane's conjecture is established, but I am unable to follow the logic. Perhaps I am missing it, but this connection seems very loose. But more importantly, I'm not sure it gives any predictive power (see comments below).

11) It is stated that the wormhole picture is verified by the data, but one could equally interpret the data as simply verifying Haldane's conjecture, while the wormhole picture is an irrelevant detail about how time-translation symmetry is explicitly broken in the replica partition function(as one would expect).

12) It is stated : “According to our picture, the ES should be similar to the energy Hamiltonian of a closed one layer system.” Again, I am not convinced of this. To me it seems like this is already an established trend, as pointed out by Haldane, and now it is being credited to the wormhole effect.

13) why is the mode near (0,0) in the bilayer so much weaker?

14) “However, Mermin-Wagner theorem [97, 98] told us that, it is impossible for a 2d system to spontaneously break a continuous symmetry at finite temperature. The only possibility is that there are long- range interactions in the HA of our system, as in the 1d cases [44], such that the Mermin-Wagner theorem is circumvented.” The way I see it, the replica geometry introduces long-range connections in imaginary time, which would give long range interactions in the corresponding D+1 dimensional stat mech model.

15) “We therefore conject[ure] that if the β is finite, the entangled edge are gapped and subsystem bulk is gapless, the low-lying ES will be more like the energy spectra of bulk. Moreover, if both bulk and edge are gapped, the β will lead to a competition of both gaps and induce a transition of the ES at finite temperature.” I find this section about amplification of the bulk energy gap to be also confusing. Moreover I don’t think this can be verified or refuted from the data that is presented.

16) In general the quality of the writing needs to be improved.

Nature Communications manuscript NCOMMS-22-38569

Title: The wormhole effect on the path integral of reduced density matrix: Unlock the mystery of energy spectrum and entanglement spectrum

Authors: Zheng Yan and Zi Yang Meng

Summary of changes:

- We have added the content of Supplementary Material into the Method section to make the paper more compact. We also changed the revised main text into the NC form.
- In response to the comments of reviewers 1, 2 and 3, we have divided the original Fig.3 into new Fig.3 and Fig.5 and added more sub-figures to help the explanations.
- In response to the comments of reviewers 1, 2 and 3, we have added more discussions and explanations about the equivalence principle and wormhole effect in the revised manuscript.
- In response to the comments of reviewers 1, 2 and 3, we have deleted the section of long range interaction of entanglement Hamiltonian, and also added the connection of our work with those pointed out by reviewer 2 in the modular Hamiltonians in the revised manuscript.
- In response to the comments of reviewers 2 and 3, we have further explained the novelty of our work in the broader context in the comparison with previous literatures. In response to the comments of reviewers, we have also explicitly pointed out the comparison of our spin ladder ES with the previous ED work.
- We have improved presentations and corrected typos, and many references are updated.

REPLY TO REVIEWER 1

Reviewer 1: *This work proposed a new approach to study low (entanglement) energy entanglement spectrum using quantum Monte Carlo method. The key insight is that entanglement spectrum can be obtained from spectral functions, when the reduced density operator is treated as the imaginary time evolution operator. This method is applied to spin models in one and two spatial dimensions. The authors provided a physical interpretation of nontrivial entanglement spectrum when two gapless systems are coupled to obtain a gapped state, which they called the wormhole effect. I think the results of this paper are quite novel and interesting, and thus would like to recommend it for publication in Nature Communication, after the following comments are addressed.*

Reply: We thank the Referee for the recommendation for publication as well as the valuable comments and suggestions, which helped us to greatly improve the presentation of our manuscript. In the following we give a point-by-point reply to these comments and have incorporated the changes in the revised manuscript.

Reviewer 1: *1. I found that the discussion of wormhole effect is not so clear. For example the authors used the term of "bulk" and "boundary" and it is not clear what they refer to, since the entanglement cut considered is between two layers, rather than across the boundary of a spatial region. I found it difficult to understand Fig 3 and the text related to it. What does the width of the arrows mean in Fig 3?*

Reply: We acknowledge the reviewer's insightful comment and suggestion. When we refer to the ladder/bilayer systems, they only have boundary without bulk after cutting the J' bonds. In the last part of our manuscript [Amplification of bulk energy gap in ES], our discussion and argument have been extended into general case in which the systems have both boundary and bulk. We note the confusions and have further revised the corresponding parts in the revised manuscript.

We have also improved our explanations about Fig.3 in the revised manuscript by splitting the wormhole effect part (Fig.3 (a) and (b) of old version) and added the new discussion of the principle of equivalence (Fig.3 (c) and (d) of old version). In the revised manuscript, for example, we have emphasized the differences between the Fig.3 (b) and Fig.5 (b). The former, thin one, means the subsystem A has only edge, and the latter, thick one, means the A has both bulk and edge.

As for the question on the width of the arrows in Fig.3, it means the strength of correlations. The thinner black line means the correlation decays more quickly, that is, the effective gap is larger. We also added more explanations about this point in the revised manuscript.

Reviewer 1: *2. Overall there are too many English errors in the draft which make it difficult for the readers. For example, in Page 4, "Furthermore, because H_A is independent on imaginary time, the rule of worldline evolution...hence the Li and Haldane's conjecture". In these three sentences there are many grammar errors and I found it difficult to guess its meaning.*

Reply: We thank the referee for for the helpful suggestions. We have further improved the English presentations in the revised manuscript, both in this particular part and in many other places.

Reviewer 1: 3. *In Page 5, the authors discuss that there is a phase transition at finite replica index n , and suggest that there must be long-range interaction in entanglement Hamiltonian. I am wondering if it is possible to directly verify the long range order by computing two-point functions for two far-away points, rather than relying on the existence of Goldstone modes.*

Reply: We thank the reviewer for this valuable suggestion. Since we can only obtain the data from integer imaginary time, at the moment, it is still difficult to confirm whether there is a phase transition or crossover between the Néel order and paramagnetic phase in the entanglement Hamiltonian. Thus, we decide to remove the discussion about the phase transition in the entanglement Hamiltonian in the revised manuscript, since we cannot eliminate the possibility of phases crossover in our present data. We are still working on this particular problem and hopefully to report the outcome in our next work.

REPLY TO REVIEWER 2

Reviewer 2: *This manuscript discusses a scheme for extracting the low-lying entanglement spectrum from world-line Quantum Monte Carlo (QMC) simulations and draws connections between the Li-Haldane conjecture and the wormhole effect in the path integral formulation of the reduced density matrix. Extracting information about the entanglement spectrum is highly relevant in quantum many-body physics but at the same time a very challenging task. The importance of this topic stems from the fact that the low-lying entanglement spectrum contains information about the topology of the system, i.e. according to the Li-Haldane conjecture, there is a one-to-one correspondence between the entanglement spectrum and the spectrum of edge excitations of the system. While the calculation of the low-lying entanglement spectrum is usually performed via wave-function-based methods, these methods show good performance for 1 dimensional systems but are difficult to apply in higher dimensions. Therefore, with the calculation of the entanglement spectrum from QMC methods and for higher-dimensional systems, this paper addresses a very relevant and timely topic.*

Reply: We thank the respected reviewer for the deep insight and precise understanding on the importance of the obtaining entanglement spectrum from QMC methods for higher-dimensional quantum many-body systems and we are fully agree with the viewpoint of the reviewer. It is our humble contribution in this work, that such a very relevant and timely and yet very challenging task, has been accomplished with clear physical picture (the wormhole effect) and controlled numerical scrutiny (QMC+SAC).

Reviewer 2: *I think the paper, even though well-written, is pretty difficult to understand. I would encourage the authors to make an effort to explain some of the basic concepts better and back up the explanation with graphical illustrations. Part of that could be done by expanding the supplemental material, which is very short. Additionally, the authors should try to explain a bit better the novelty of their method with respect to other proposals for extracting the entanglement spectrum in QMC simulations. Is this really the first time this has been done? If these points are addressed convincingly, I would be supportive of publication in Nature Communications. I will give more details below.*

Reply: We thank the reviewer for the valuable suggestions. Indeed, we now tried to greatly expand the content and completely redraw the Figures 3 and 5 in the revised main text, to better explain our basic concepts. Also, we have tried our best to explain the novelty of our work with respect to other proposals. In the following, we reply point-to-point to the questions raised.

Reviewer 2: *The paper can very clearly be separated into 2 parts: The first part discusses the method for obtaining the low-lying entanglement spectrum from imaginary time correlation functions and the replica partition function of the reduced density matrix. The second part tries to explain the Li-Haldane conjecture via a wormhole picture in the worldline evolution of the replica system.*

Reply: We have now restructured our revised manuscript according to this suggestion.

Reviewer 2: *The first part sounds to me like a good and pretty obvious idea. Since the spectrum in MC*

*is evaluated from correlation functions in imaginary time, it sounds quite natural to me to introduce many copies of the system such as to mimic imaginary time evolution with the modular Hamiltonian H_A at integer times. One actually wonders why this idea was not pursued earlier in the QMC community. I have to admit that I do not really understand the difference (the novelty) with respect to section III in the paper Phys. Rev. B **89**, 125121. The authors should try to explain better how their method is related to that as well as to other approaches of extracting the entanglement spectrum in QMC. Other than that, the results look convincing. I would suggest adding to Figure 4 c) the exact results for $L = 14$ which show a linear dispersion and contrast it with the quadratic dispersion which is revealed from the QMC simulations on larger system sizes.*

Reply: We thank the respected reviewer for raising the insightful comments and suggestions. Although to introduce many replicas of the systems and to compute the imaginary evolution might sounds obvious, there were actually very few previous QMC works along the line, and even for these previous works, the physical picture and the numerical results were not clearly presented, partially because the lack of clear guidance of physical picture and the increasement of the configuration space for sampling and the consequent reduced data quality. In this regards, we believe our work is the first to clearly achieve both the conceptual and numerical presentation such that this simple idea can be convincingly stated and easy for a wider audience to appreciate. We understand the curiosity of the reviewer that why such work has not be previously carried out, but we would say that such hindsight could also apply to many other discoveries in science that once understood things look simple and straightforward.

In the revised manuscript, we have explained more about the relation between our approach and the works like Phys. Rev. B **89**, 125121, and let us also address it fully here.

The difference of our work and previous ones such as Phys. Rev. B **89**, 125121 are at least two-folded.

First, at the conceptual level, previous works such as the one mentioned, did not give rise to the clear path-integral interpretation in the wormhole effect of world-line of the entanglement spectrum in the replicated space-time manifold, without which, we believe cannot fully explain the obtained entanglement spectrum in the context of the Li-Haldane conjecture. And it is precisely in such context, we believe they do not give the conceputal breakthrough in understanding the entanglement spectrum from QMC for quantum many-body systems as we do.

Second, also related with the first point, is that at the technical level, although the auxiliary field QMC paper and our work both computed the entanglement spectrum, our work is the first one in doing so in boson/spin QMC. And we think it is because the relative complicated mathematical derivation in the auxiliary field QMC to translate an interacting fermionic partition function into sampling of determinants, that the previous work did not fully convey the message of computing the entanglement spectrum in the replicated manifold in QMC and many people who are not familiar with the auxiliary field QMC didn't fully appreciate the idea. In contrast, we believe our path-integral interpretation and the world-line approaches greatly simplified the key physical idea of the computation of entanglement spectrum in QMC simulations with much large system sizes 50×50 in 2D and much better data quality and humbly think that our presentation will make such computation more accessible to the broader audience.

As for the suggestion about the Fig 4 (c), we are a bit confused here, does the respected reviewer mean the result of exact diagonalization (ED)? The $L = 14$ means the bilayer should be about $14 \times 14 \times 2 =$

392 spins which is too large for ED. We think the respected reviewer may mean the Fig.2 (c). In fact, the linear fitting and related ES in small size via ED have been done in the D. Poilblanc's PRL paper (Phys. Rev. Lett. 105, 077202 (2010)). We thank the respected reviewer for the good suggestion and have willingly added the detailed figure number of the ED fitting result of that reference in our revised manuscript to help comparisons expediently and highlight the severe finite size effect with ED in obtaining the entanglement spectrum. It is important to have the QMC results like ours to clearly reveal the true entanglement information.

FIG. 4 (color online). (e),(f) The same as Fig. 3 for the rung singlet (II) phase (only $L = 10$ and $L = 14$ are shown). Here the GS is the saturated ferromagnet. The total spins S of the lowest eigenstates are indicated by different symbols (and colors) and can be assigned to m -magnon bound states, $m = S_{\max} - S = L/2 - S$. The lowest-energy excitations for $L = 14$ are fitted according to $E_{\min}(K)$ (see text). In (e), the fit for $L = 14$ rescaled by a factor $14/10$ (upper dotted line) also gives good agreement with the $L = 10$ data.

FIG. R1: The Fig.4 of D. Poilblanc's PRL paper. The left one ($\theta = 2\pi/3$) is same parameter as the Fig.2(c) of our paper. The fitting line is linear around the gapless point.

Reviewer 2: *I have to admit that (even after reading the section several times) I do not completely understand the wormhole picture the authors try to convey. I would suggest the authors make an effort trying to explain this better, maybe with a dedicated section in the supplemental material. Figure 3 (a), (c), and (d) are confusing to me. It is not entirely clear what's on the axis and maybe one should really expand this picture in the Supplemental Material instead of just writing "from another replica" and "to another replica".*

Reply: We thank the reviewer for the valuable suggestions. We completely agree with the reviewer that the description in this part is not clear enough. In the revised version of the manuscript, we have now divided this part into two parts (as suggested by the reviewer) with more explanations and added more figures to help readers' understanding. At first, we split the content into "equivalence principle" and "wormhole effect" two sections. The Fig.3 explains the similarities (equivalence) of the world-line evolutions of the closed system and the replica system as your understanding in next comment below. The Fig.5 describes the wormhole effect. Through the Fig.5 (b), it is much easier to understand the worldline choices with different

cost in Fig.5 (a). Furthermore, the red/blue paths in Fig5. (b) are much more clear to show the lengths of the bulk/edge paths are greatly different. We hope these efforts will help the readers to understand the edge path is much shorter with fewer action cost in the path integral, thus, the edge mode is more important in the low-lying ES.

*Reviewer 2: If I understand correctly, from similarities (equivalence) of the world-line evolutions of the closed system and the replica system, the authors conclude that \mathcal{H} is similar to \mathcal{H}_A . I would like to make the authors aware that the modular Hamiltonian for ground states has been investigated in several works (see e.g. *Nature Physics* volume **14**, pages 827-831 (2018), *Phys. Rev. B* **98**, 134403 or *Phys. Rev. B* **100**, 155122). Here it is shown that the modular Hamiltonian has a similar "local" structure as the system Hamiltonian, but different coupling coefficients which in a way define a local "entanglement temperature". Would it be possible to confirm this picture in QMC simulations? Would it make sense to compare the world-line evolution of such a "deformed" modular Hamiltonian with the one of the system Hamiltonian?*

Reply: We thank the reviewer for the excellent comment and references. These papers are highly related with our work, so we have cited them in the revised version. In fact, we also noted the point of "entanglement temperature" when completed this work, that the entanglement Hamiltonian is similar with the system Hamiltonian but experiences a different "entanglement temperature" as we tune the couplings between the subsystem and the environment. We are now looking into more details on the form of entanglement (modular) Hamiltonian and "entanglement temperature", hopefully we could report results along this line in future work.

Our rough understanding for now is, from the Fig.4 of our manuscript, it seems the entanglement Hamiltonian (EH) doesn't change when we tune the coupling J' , but the phase of the EH under different J' changes. When the J' increases, the phase changes from ordered (Neél) to disordered (dimerized). Why the same EH will induce different phases? The answer is the effective temperature of the EH changes from low temperature to high. We think this is also what the respected reviewer meant by defining a local "entanglement temperature". However, our following work is still on-going and there are still parts not completely clear both in simulation and in data, thus we hope the discussion of this part will be kept in our coming paper. We hope the reviewer could understand our choice.

*Reviewer 2: Related to the above: In recent work, it has been shown that parent Hamiltonians of Gibbs states can be reconstructed from local observables on the system (see e.g. *Quantum* **3**, 159 (2019)). Given the above that the modular Hamiltonian exhibits a local structure, these methods could be applied to extract the entanglement Hamiltonian from local observables measured in QMC simulations. Would it be conceivable to extract the entanglement spectrum from imaginary time evolution with the "learned" modular Hamiltonian? What do the authors think of how such a method would compare to their proposal? Would it make sense to add a comment on this in the paper?*

Reply: We thank the reviewer for the excellent question and references. Certainly, we have added such a comment in our revised manuscript.

Indeed, the question and suggestions are very relevant. We have read the *Quantum* **3**, 159 (2019) paper, and think it is a little hard for QMC at least, but the concept can certainly be discussed, especially about

how to efficiently and reliably extract the "learned" entanglement (modular) Hamiltonian. The key point is to recontract the Hamiltonian via the correlation matrix of the state. We have found similar work using this method, W. Zhu, Zhoushen Huang, and Yin-Chen He, "Reconstructing entanglement hamiltonian via entanglement eigenstates", Phys. Rev. B 99, 235109 (2019). It seems possible for numeric studies. However, for QMC method, it is hard to obtain the correlation matrix because the measurement is not universal. It is easy for QMC to extract the diagonal correlations but not off-diagonal ones. In addition, the computer memory decides the size of the correlation matrix and further limits the EH size. In any case, there relevant discussion has been added in the revised manuscript.

REPLY TO REVIEWER 3

Reviewer 3: *The authors present a study of the entanglement spectrum of quantum spin models via large-scale quantum Monte Carlo simulations in one and two dimensions. The results are significant, since the system sizes studied are an order of magnitude larger than previous studies, and the first computation of the entanglement spectrum for a 2D entangling region is presented. The results support the conjecture by Haldane that the entanglement spectrum has a close correspondence with the energy spectrum of the subsystem in isolation.*

The results have been obtained by a novel combination of existing cutting edge techniques, namely the replica time-displaced correlation function introduced in ref [41], and the stochastic analytic continuation method. The resulting data is scientifically sound with a high degree of confidence.

Reply: We thank the respected reviewer for the concise summary and positive assessment of our work. These insightful comments of the reviewer, have further inspired us to improve the manuscript, we reply them one by one now and have made the correspondingly changes in the revised manuscript.

Reviewer 3: *A main focus of the paper is on the so-called "wormhole effect" whereby correlations decay exponentially in imaginary time, but experience revivals due to the presence of replicas. Though this is an interesting effect which is being presented for the first time, is not clear that this observation impacts the results or conclusions of the paper in any way. Indeed ref [41] studied the replica time-displaced correlation function only at integer time displacements (only at the revival peaks), representing integer powers of the reduced density matrix. As such I don't believe this effect should be a centerpiece of this paper, and I think naming it the wormhole effect is potentially misleading.*

Overall, I think that this is a significant numerical study of entanglement spectra in large-scale quantum Monte Carlo simulations, which would be prohibitive with other methods. However, I find that the impressive results have been overshadowed by an unnecessary emphasis on a curious, but not surprising, effect with a sensationalized name (the wormhole effect). I would recommend publication in nature communications provided that the emphasis in the paper be significantly redirected to focus on the results and the novel implementation of the methods. Below I will give a list of suggested improvements.

1) I think the title is too sensational, and I haven't been convinced that the wormhole effect explains Haldane's conjecture.

Reply: We thank the reviewer for the kind comments and suggestions. We have taken them very seriously and would like to clarify with the respected reviewer, that although we thank the reviewer deeply for his/her appreciation of our numerical implementation of the method (QMC+SAC) and results such as the first computation of the entanglement spectrum for 2D entangling region, and our results support the Li-Haldane conjecture of the entanglement spectrum, but we do believe our wormhole effect picture is also important in that, it not only explains the Li-Haldane conjecture, but further extends the conjecture into more general cases. We think this is the key point of this work which is equally important as the cutting edge numerical results.

We have noticed that it is very likely due to our poor presentation that the wormhole effect is not clearly

stated to the respected reviewer. To clarify such unfortunate misunderstanding, we have rewritten the corresponding parts in the revised manuscript and divided the Fig.3 of main text into two parts: equivalence principle and wormhole effect. We would also like to further explain the wormhole effect to the reviewer below, and hopefully could convey the message clearly.

In fact, we think the wormhole effect on path integral is a direct and deep picture to reveal the mechanism of entanglement spectra (ES), according to which, we further successfully realized the "Reversing the Li and Haldane conjecture: The low-lying entanglement spectrum can also resemble the bulk energy spectrum" via tuning the temperature (length of imaginary time) in our followup work (arXiv: 2210.10062). Basically, it not only explains the Li and Haldane conjecture but also guide us to predict new phenomena and to reverse it.

As shown in Fig. R2, panel (a) is the same to the Fig.1 in present manuscript which explains how to translate the evolution of reduced density matrix (RDM) into the replicated configuration of QMC, and panel (b) uses two paths to explain the wormhole effect with simplified abstraction, that, the yellow one is deeply in the bulk of A and the blue one is around the entangled edge between A and \bar{A} . One sees the imaginary time length of the yellow path is much longer than that of the blue one. The trace of the \bar{A} means the upper and lower boundaries of every replica are connected, as shown by the dashed lines in the Fig.R2 (a). This means the distance between the upper and lower boundaries of \bar{A} is very short (at the scale $\sim \beta$) while the distance between the upper and lower boundaries of A is as long as the scale $\sim n\beta$. Based on path-integral, the importance of a path takes the weight $e^{-\int \mathcal{D}(\mathcal{L})\Delta(\mathcal{L})}$, where the \mathcal{L} is the path and $\Delta(\mathcal{L})$ is the gap of this path, which means the path with shorter path/smaller gap plays more important role among all the paths to contribute to the partition function. In the zero temperature $\beta \rightarrow \infty$, the path around the entangled edge (the blue line) is more important, as the integral $\int \mathcal{D}(\mathcal{L})\Delta(\mathcal{L})$ of this path is closer to zero and give rise to larger weight than that of the yellow path deep inside the bulk. Thus, the ES is always reminiscent of the edge spectrum, and this is the interpretation of the Li and Haldane conjecture in the wormhole language.

Following this understanding, we predict that the low-lying entanglement spectrum can also resemble the bulk energy spectrum when the temperature is higher (shorten the yellow path) and the edge gap is larger than bulk (increase the edge gap). The prediction has been successfully demonstrated in our new article (arXiv: 2210.10062), in which, we have both realized the Li and Haldane conjecture at low temperature and more importantly for the present discussion, reversed it by engineering the replicated manifold and the relative strength of the bulk and edge gaps.

We have tried our best to incorporate these explanations and improvement of our presentations of the wormhole effect in the revised manuscript, and we hope the respected reviewer could now appreciate our conceptual breakthrough in this work, along with our numerical results.

Reviewer 3: 2) For a balanced view, I think it is important to mention references that are critical of the supposed universality of the entanglement spectrum, for instance Phys. Rev. Lett. 113, 060501.

Reply: We thank the respected reviewer for the insightful suggestion, indeed, it is important to add the critical reference on the universality of the entanglement spectrum, and we have willingly done so in the revised manuscript.

FIG. 1. (a) Graphical representation of the partition function $\mathcal{Z}_A^{(n)}$ in the replicated manifold. The shaded area is the entanglement region A in which all the replicas are glued together along the imaginary time and has length $\beta_A = n\beta$. In the environment region \bar{A} , replicas are independent along the imaginary time axis and each has length β . (b) Two different worldline paths along the imaginary time. Yellow one travels inside the bulk while blue one goes into the environment and experiences the wormhole effect. Black circles are the wormholes which teleport a worldline to the other side of a replica in \bar{A} via the virtual path (blue dashed line).

FIG. R2: The Fig.1 of our new work (arXiv: 2210.10062).

Reviewer 3: 3) *In the introduction, general statements about the Monte Carlo method are made, but the supporting citations are specific studies by one of the authors (ref [49] and [50]), which don't seem to fit.*

Reply: Thanks for the respected reviewer's warm tips. We have made suitable modifications by citing more appropriate references here.

Reviewer 3: 4) *From the introduction: "Here we first develop a protocol to overcome the exponential growth of computational complexity in obtaining the ES via QMC combined with stochastic analytic continuation." As far as I know, this technique has already existed in ref [41] and it was pointed out in ref [42] that it could be used for stochastic series expansion Monte Carlo for spin systems as done in this work.*

Reply: We thank the reviewer's kind reminder. We now have read these papers more carefully and revised our presentation to further clarify the novelty of our work. However, we do think there are clear difference between our work and these references, and would like to explain below to the respected reviewer.

First, at the conceptual level, previous works such as the ones mentioned, did not give rise to the clear path-integral interpretation in the wormhole effect of world-line of the entanglement spectrum in the replicated space-time manifold, without which, we believe cannot fully explain the obtained entanglement spectrum in the context of the Li-Haldane conjecture. And it is precisely in such context, we believe they do not give the conceptual breakthrough in understanding the entanglement spectrum from QMC for quantum

many-body systems as we do.

Second, also related with the first point, is that at the technical level, although the auxiliary field QMC paper and our work both computed the entanglement spectrum, our work is the first one in doing so in boson/spin QMC and the first one which truly computes the 2D entangled boundaries (in the previous works, the entanglement boundaries are still in the ribbon geometry and therefore 1D in construction). And we think it is because the relative complicated mathematical derivation in the auxiliary field QMC to translate interacting fermionic partition function into sampling of determinants, that the previous work did not fully convey the message of computing the entanglement spectrum in the replicated manifold in QMC and many people who are not familiar with the auxiliary field QMC didn't fully appreciate the idea. In contrast, we believe our path-integral interpretation and the world-line approaches greatly simplified the key physical idea of the computation of entanglement spectrum in QMC simulations with much large system sizes 50×50 in 2D and much better data quality and humbly think that our presentation will make such computation more accessible to the broader audience.

Reviewer 3: 5) It is frequently stated that there are comparisons with exact diagonalization (ED) from reference [27], although this data is not included in the main text and is not clear how to compare with the given reference. This data should be presented in the main plot. Moreover, I think it is very important to give direct comparisons with exact results on small system sizes. Can't one use the QMC technique on $L=10$ and compare with exact results directly?

Reply: Thanks a lot for the good suggestions to improve the readability of our manuscript. In the revised manuscript, we have added more details to guide the comparisons (e.g., list the corresponding figure number of the ED paper and explain the parameters used in the comparison). But we are sorry to say that it will actually be difficult to add the small size QMC results. This is because we can only obtain the integer imaginary time correlations of the entanglement Hamiltonian (EH) to support the SAC to generate good quality spectra, which means we can only extract the low-lying ES, but the finite size effect of the small size spectrum make the gaps much higher and renders both the poorer spectrum and very few momenta can be used. We hope the respected reviewer could understand the difficulty.

Reviewer 3: 6) Many branches of the spectra cannot be seen by eye without zooming in as much as possible on a pdf. Perhaps changing the color scale would make these stand out?

Reply: We thank the reviewer for the kind suggestions to help us improve the presentation. In fact, we also noticed this problem and tried to change the color scale (e.g., log scale) to show the ES, but the result is not good. The reason is that we are limited to the integer imaginary time correlations, it is challenging for SAC to generate better spectra compared with the continuous time correlation functions. There are some small noises if we use the log scale to draw the ES. That's why we have added the leader lines to point out the weak dispersions in our figures. To make the weak mode near (0,0) more clear, we have set the width of Fig.4 doubled.

Reviewer 3: 7) Velocities are extracted, do these agree with theoretical predictions? This fitting for ED is not shown.

Reply: Thanks very much for the question. Since the velocities are those of the entanglement Hamiltonian, whose exact form are still unknown, it is hard to precisely pin down the value of the velocities, but our obtained function form agree with the theoretical prediction. We have added the corresponding figure number of the ED reference paper to help the comparison.

Reviewer 3: 8) From wormholes in the path integral section: "one can use a wormhole effect of the world-line in QMC simulation in space-time to explain the tracing of the environment in the path-integral." The logic here is reversed. The tracing of the environment explains the the wormhole effect.

Reply: Yes, we thank the respected reviewer for the careful reading. Indeed, your logic is right and we have modified this sentence according to your suggestion.

Reviewer 3: 9) I find figure 3c and 3d a bit confusing. I think the point that worldlines can make shortcuts through imaginary time by travelling through replicas is well explained already from fig 3a.

Reply: We thank the respected reviewer for the insight comment. Indeed, we now rewrite this part with more figures and explanations, to be consistent with the suggestion of the respected reviewer and that of the reviewer 2: "from similarities (equivalence) of the world-line evolutions of the closed system and the replica system". Our new presentation deliver the message that the physical laws (operators and world-lines of entanglement Hamiltonian) measured from the integer time points of replica system, is similar to the physical laws of the edge Hamiltonian we can see in a closed system. That's why the entanglement Hamiltonian is similar to the edge Hamiltonian. We hope our new presentation could be satisfactory for the respected reviewer.

Reviewer 3: 10) I find the second paragraph of the "wormholes in the path integrals" section particularly confusing. Presumably this is where the connection between the wormhole effect and Haldane's conjecture is established, but I am unable to follow the logic. Perhaps I am missing it, but this connection seems very loose. But more importantly, I'm not sure it gives any predictive power (see comments below).

Reply: We thank the reviewer for the suggestion. Indeed, we have now completely revised this section and tried our best to establish the connection between the wormhole effect and the Li-Haldance conjecture in the revised manuscript.

As we have mentioned above, the wormhole effect can predict new phenomena and we have done so in the our new work (arXiv: 2210.10062), where the ES can be reversed to resemble that of the bulk energy spectrum.

Reviewer 3: 11) It is stated that the wormhole picture is verified by the data, but one could equally interpret the data as simply verifying Haldane's conjecture, while the wormhole picture is an irrelevant detail about how time-translation symmetry is explicitly broken in the replica partition function(as one would expect).

Reply: We thank the respected reviewer for the good comment. As we have explained above, our data certainly verify the Li-Haldane conjecture, and we believe our wormhole effect offers the simple yet fun-

damental picture to explain the working of the conjecture. More importantly, the wormhole effect has the predicting power to reverse the Li-Haldane conjecture such that one can engineer situation where the ES resembles that of the bulk energy spectrum, as we have successfully done so in our new work k (arXiv: 2210.10062).

Reviewer 3: 12) It is stated : "According to our picture, the ES should be similar to the energy Hamiltonian of a closed one layer system." Again, I am not convinced of this. To me it seems like this is already an established trend, as pointed out by Haldane, and now it is being credited to the wormhole effect.

Reply: We are sorry for our poor descriptions. We have revised our description in the revised manuscript.

Reviewer 3: 13) why is the mode near (0,0) in the bilayer so much weaker?

Reply: Thanks for the good question. Because the ES is similar to the energy spectrum of single layer antiferromagnetic Heisenberg model, the ground state of the entanglement Hamiltonian should be a Néel order (condensed at (π, π)), it is well-known that in such situation, the spectrum weight near the (π, π) is much higher than others, especially near (0,0). Similar results can be seen in many references such as Fig.5 in (Phys. Rev. X 7, 041072) or Fig. 2(a) in (Phys. Rev. Lett. 126, 227201), to name a few.

Reviewer 3: 14) "However, Mermin-Wagner theorem [97, 98] told us that, it is impossible for a 2d system to spontaneously break a continuous symmetry at finite temperature. The only possibility is that there are long-range interactions in the HA of our system, as in the 1d cases [44], such that the Mermin-Wagner theorem is circumvented." The way I see it, the replica geometry introduces long-range connections in imaginary time, which would give long range interactions in the corresponding D+1 dimensional stat mech model.

Reply: We thank the respected reviewer for the good argument. We certainly agree this argument, however, since we can only obtain the data from integer imaginary time, at the moment, it is still difficult to confirm whether there is a phase transition or crossover between the Néel order and paramagnetic phase in the entanglement Hamiltonian. Thus, we decide to remove the discussion about the phase transition in the entanglement Hamiltonian in the revised manuscript, because we cannot eliminate the possibility of phases crossover in our present data. We are still working on this particular problem and hopefully to report the outcome in our next work.

Reviewer 3: 15) "We therefore conjecture that if the β is finite, the entangled edge are gapped and subsystem bulk is gapless, the low-lying ES will be more like the energy spectra of bulk. Moreover, if both bulk and edge are gapped, the SA will lead to a competition of both gaps and induce a transition of the ES at finite temperature." I find this section about amplification of the bulk energy gap to be also confusing. Moreover I don't think this can be verified or refuted from the data that is presented.

Reply: We thank the reviewer for the careful reading and good suggestion. As we mentioned above, we realized the aforementioned statement in our new article (arXiv: 2210.10062). It strongly supports our conjecture and make us much believe the wormhole effect is the basic mechanism of ES.

Reviewer 3: *16) In general the quality of the writing needs to be improved.*

Reply: Yes, we completely agree with the respected reviewer. In the revised manuscript, we have tried our best to improved the readability, and have also rewritten many parts to improvement the presentation, in accordance with the suggestions of all three reviewers. We sincerely thank the respected review for the helpful comments and suggestions.

REVIEWERS' COMMENTS

Reviewer #3 (Remarks to the Author):

Final comments:

The authors have gone to great lengths to improve explanations of the wormhole effect and why it underpins the Li-Haldane conjecture. I am now convinced that this is an important physical principle that is worthy of its name and its central role in the paper. Indeed, this picture has predictive power, as the authors have demonstrated here and in another recent publication. I believe the paper should be published, and I give an optional suggestion for modification below and some grammar suggestions at the end.

Suggestion:

I appreciate the expanded Fig 3. and supporting text in the new "Equivalence principle of ES" section, however I do find it to be a bit confusing. Doesn't the "Wormholes in the path integral" section serve the same function? Specifically, I thought the wormholes were the key to explaining Li-Haldane's conjecture, and yet they are not depicted in Fig. 3 and are not discussed in the equivalence principle section.

I do think that the "Wormholes in the path integral" section has been greatly improved and is now very convincing. My suggestion would be to use this section in place of the "Equivalence principle" section, and maybe remove the equivalence principle section entirely. For instance, with Fig. 3 included, I think there are just too many pictures of replica partition functions.

Grammar Suggestions:

This is optional, but I feel that the title would read better if unlock -> unlocking

abstract:

- Heisenberg spin ladder -> the Heisenberg spin ladder

Intro P1:

- categorical -> categorical

Intro P2:

- exponentially growth -> exponential growth

- important sampling -> importance sampling

Intro P3:

- is indeed behave -> does indeed behave

- the Haldane's conjecture -> Haldane's conjecture (everywhere)

- will determine the behavior of low-lying ES -> that will determine the behavior of low-lying ES

Section title "Result" -> Results

Results P1:

- As known in closed system, spectral function $S(\omega)$ for physical observable -> As known in closed systems, the spectral function $S(\omega)$ for physical observables

Results P2 :

- periodical boundary condition -> periodic boundary condition

- We note similar approach -> We note a similar approach

Spin-1/2 Ladder P1:

- ES of Heisenberg ladder -> ES of the Heisenberg ladder

- The spins on ladder are coupled through the nearest neighbor Heisenberg interactions -> The spins on the ladder are coupled through nearest neighbor Heisenberg interactions

Spin-1/2 Ladder P3:

- Since the ES in TDL is expected to show spectrum -> Since the ES in TDL is expected to show the spectrum

Fig 3 caption:

- which is same as the Fig.1 -> which is the same as Fig.1.

Equivalence Principle P1:

- spectrum of subsystem within the frame -> spectrum of the subsystem within the frame

Equivalence Principle P2:

- space size of the entanglement Hamiltonian seems as same as the subsystem A, but we haven't explain -> spatial size of the entanglement Hamiltonian is the same as the subsystem A, but we haven't explained

Equivalence Principle P3:

- With above picture -> With the above picture

P4:

- Reader may ask -> The reader may ask

Nature Communications manuscript NCOMMS-22-38569

Title: The wormhole effect on the path integral of reduced density matrix: Unlock the mystery of energy spectrum and entanglement spectrum

Authors: Zheng Yan and Zi Yang Meng

Summary of changes:

- Considering the reviewer's suggestion in last round, we have changed the title to make a balance between the wormhole effect and numerical method.
- We have corrected some typos and grammars according to the reviewer's suggestion, thus we haven't highlighted them with color.

REPLY TO THE REVIEWER

Reviewer 3: *Final comments:*

The authors have gone to great lengths to improve explanations of the wormhole effect and why it underpins the Li-Haldane conjecture. I am now convinced that this is an important physical principle that is worthy of its name and its central role in the paper. Indeed, this picture has predictive power, as the authors have demonstrated here and in another recent publication. I believe the paper should be published, and I give an optional suggestion for modification below and some grammar suggestions at the end.

Suggestion:

I appreciate the expanded Fig 3. and supporting text in the new Equivalence principle of ES section, however I do find it to be a bit confusing. Doesnt the Wormholes in the path integral section serve the same function? Specifically, I thought the wormholes were the key to explaining Li-Haldanes conjecture, and yet they are not depicted in Fig. 3 and are not discussed in the equivalence principle section.

I do think that the Wormholes in the path integral section has been greatly improved and is now very convincing. My suggestion would be to use this section in place of the Equivalence principle section, and maybe remove the equivalence principle section entirely. For instance, with Fig. 3 included, I think there are just too many pictures of replica partition functions.

Reply: We thank the Referee for the recommendation for publication as well as the valuable suggestions. For the Equivalence principle of ES section, we think it is complementary to the Wormholes in the path integral section. Because the Equivalence principle of ES answers why the entanglement spectrum (ES) resembles energy spectrum when there is only edge and no bulk, while the Wormholes in the path integral answers why the ES is much like the edge energy spectrum when there are both bulk and edge in the system. According to the respected reviewer thought this is an "optional suggestion", we decide to remain this important part. Thanks very much for the good suggestion all the same.

Reviewer 3: *Grammar Suggestions:*

This is optional, but I feel that the title would read better if unlock -¿ unlocking

abstract: - Heisenberg spin ladder -¿ the Heisenberg spin ladder

Intro P1: - categorical -¿ categorical

Intro P2: - exponentially growth -¿ exponential growth - important sampling -¿ importance sampling

Intro P3: - is indeed behave -¿ does indeed behave - the Haldanes conjecture -¿ Haldanes conjecture (everywhere) - will determine the behavior of low-lying ES -¿ that will determine the behavior of low-lying ES

Section title Result -¿ Results

Results P1: - As known in closed system, spectral function $S()$ for physical observable -¿ As known in closed systems, the spectral function $S()$ for physical observables

Results P2 : - periodical boundary condition -¿ periodic boundary condition - We note similar approach -¿ We note a similar approach

Spin-1/2 Ladder P1: - ES of Heisenberg ladder -¿ ES of the Heisenberg ladder - The spins on ladder

are coupled through the nearest neighbor Heisenberg interactions -ζ The spins on the ladder are coupled through nearest neighbor Heisenberg interactions

Spin-1/2 Ladder P3: - Since the ES in TDL is expected to show spectrum -ζ Since the ES in TDL is expected to show the spectrum

Fig 3 caption: - which is same as the Fig.1 -ζ which is the same as Fig.1.

Equivalence Principle P1: - spectrum of subsystem within the frame -ζ spectrum of the subsystem within the frame

Equivalence Principle P2: - space size of the entanglement Hamiltonian seems as same as the subsystem A, but we havent explain -ζ spatial size of the entanglement Hamiltonian is the same as the subsystem A, but we havent explained

Equivalence Principle P3: - With above picture -ζ With the above picture

P4: - Reader may ask -ζ The reader may ask

Reply: We thank the reviewer for the careful reading and helpful suggestions. We have corrected all of these. We sincerely thank the respected reviewer's recommendation and helps.